# Is There a Difference between Perineural Dexamethasone with Single-Shot Interscalene Block (SSIB) and Interscalene Indwelling Catheter Analgesia (IICA) for Early Pain after Arthroscopic Rotator Cuff Repair? A Pilot Study

**DOI:** 10.3390/jcm11123409

**Published:** 2022-06-14

**Authors:** Yang-Soo Kim, Youngkyung Park, Hyun Jung Koh

**Affiliations:** 1Department of Orthopedic Surgery, College of Medicine, The Catholic University of Korea, Seoul 06591, Korea; kysoos@catholic.ac.kr; 2Department of Anesthesiology and Pain Medicine, College of Medicine, The Catholic University of Korea, Seoul 06591, Korea; dennypark21@gmail.com

**Keywords:** arthroscopy, analgesic, dexamethasone, interscalene block, perineural, rotator cuff repair

## Abstract

Interscalene block is applied to control acute postoperative pain after arthroscopic rotator cuff repair (ARCR), typically with single-shot interscalene block (SSIB) or continuous interscalene indwelling catheter analgesia (IICA), and dexamethasone (Dex) for extending the analgesic effect. This study investigated whether perineural Dex can extend the postoperative analgesic effect of SSIB to match that of IICA. A total of 130 patients were recruited and divided into two groups (Group D, SSIB with perineural Dex, *n* = 94; Group C, IICA, *n* = 36). The surgical and anesthetic processes were identical except for the method of nerve block. Pain was measured by a visual analog scale (VAS) at 6, 12, 24, and 48 h after ARCR. The number of each and the total analgesics used and adverse effects were compared. The duration of ARCR was longer in group D. The VAS score was higher in group C 6 h after ARCR, but there was no difference at other time points. More postoperative analgesics were administered to group C, and there was no difference in the number of adverse effects. In conclusion, combining perineural Dex with SSIB can reduce rebound hyperalgesia after 6 h and extend the duration of the analgesic effect to that of IICA. Therefore, IICA could be substituted with SSIB and Dex between at 6 and 48 h after ARCR.

## 1. Introduction

Various methods of interscalene block can be applied to control acute postoperative pain after arthroscopic rotator cuff repair (ARCR). The most common of these are single-shot interscalene block (SSIB) and continuous interscalene indwelling catheter analgesia (IICA). Both methods have their advantages and disadvantages: SSIB, for example, is more effective for relieving the acute pain that occurs immediately after ARCR due to injection of a high dose of analgesic at once rather than via a continuous infusion, but when the effect of local anesthesia diminishes, the pain becomes more severe. Such rebound hyperalgesia results in the administration of more analgesics, and adverse events more easily occur.

In contrast to SSIB, rebound hyperalgesia does not occur in IICA due to the continuous nature of the infusion [1,2,3]. However, problems associated with IICA include the difficulty in relieving pain during very early periods after ARCR, the discomfort of catheter itself, neurological problems, and the inconvenience of managing the patient-controlled analgesia (PCA) device. In such cases, the selection of analgesic methods may be considered according to the surgeon’s preference and the patient’s condition. In addition, as the number of outpatient surgeries at the day care center (DCC) continues to increase, the length of hospital stay is becoming shorter, and the return to daily life is accelerating. For these reasons, SSIB is preferred to IICA. However, to maximize the effect of SSIB, the rebound hyperalgesia must be reduced or prevented. One solution previously described in the literature is prolongation of the analgesic effect via the addition of various adjuvants [4,5,6,7,8]. Among them, dexamethasone (Dex) has been shown to prolong the duration of the postoperative analgesic effect and lessen rebound hyperalgesia [9]. Studies have investigated the effect of dexamethasone on the prolongation of the analgesic effect from a number of different perspectives [10,11,12,13,14]. However, there were no studies for investigating in regard to the effect of the use of Dex as an adjuvant for SSIB versus IICA on pain control more than 6 h after ARCR among research on nerve block type and duration of analgesic effect. Therefore, a comparison of the two nerve blocks after this time point would demonstrate the importance of choosing the best postoperative analgesia. Accordingly, the aim of this study was to investigate whether SSIB combined with Dex could replace IICA according to different outcome parameters during the early phase of the postoperative period after ARCR.

## 2. Materials and Methods

### 2.1. Study Design

We recruited patients with rotator cuff tears measuring less than 2 cm who had visited Seoul St. Mary’s Hospital from March 2018 to March 2020. The Ethical Committee Institutional Review Board (IRB) approved this study (Ethical Committee N° KC17OESI0118), which was registered to the Clinical Research Information Service (CRIS) (registration number (KCT0007119). Informed consent was obtained from the patients for participation in this study.

### 2.2. Participants

Patients with American Society of Anesthesiologists physical status (ASA—PS) I-III were enrolled to undergo elective ARCR. The exclusion criteria were age below 19 years, severe renal or hepatic insufficiency, severe cardiac disease, epilepsy, dementia, cognitive dysfunction, cerebrovascular disease, allergies or contraindications to local anesthetics or opioids, and a skin infection or wound in the block site. Patients who could not understand the visual analog scale (VAS) or were unable to understand how to operate the interscalene-PCA were also excluded.

### 2.3. Patient Allocation

This study was conducted according to the following operation process. A total of 130 patients were recruited and included in one of two groups during two years: group D included patients who underwent SSIB with perineural Dex (*n* = 94), and group C included patients who underwent IICA (*n* = 36). The nerve block type was decided by surgical order: the first operation implemented SSIB with perineural Dex, the next operation implemented IICA, and so on. Approximately three times as many patients underwent the first operation as underwent the second operation.

### 2.4. Procedure and Intervention

#### 2.4.1. Operative Techniques

All patients underwent ARCR while in the semilateral decubitus position under general anesthesia. A single orthopedic surgeon performed all surgical procedures. Arthroscopic single-row repair was performed for patients with small rotator cuff tears, while medium tears were repaired with the double-row repair technique. The double-row bridge technique was used in most cases; however, in cases of tense tendons with poor mobility even after adequate release, we used single-row repair using the Mason–Allen technique.

#### 2.4.2. Anesthetic Methods

All operations were performed in the same manner except for the postoperative analgesia delivery. General anesthesia was induced by propofol 1.5 mg/kg and rocuronium 0.6 mg/kg and maintained with desflurane 4~6 vol%. After ARCR, patients in group D underwent SSIB with 0.45% ropivacaine 12 cc, 5% dextrose 7 cc and dexamethasone 5 mg. Patients in group C underwent IICA; a StimuLong interscalene catheter (PAJUNK GmbH, Geisingen, Germany) was placed at the level of the C5/6 brachial plexus via ultrasound guidance. Catheter placement was confirmed by a nerve stimulator (B. Braun, Melsungen, Germany), after which the catheter tip was placed near the C5 and C6 nerve roots at a depth of 3 to 5 cm. Immediately after ARCR, 10 mL bolus (0.3% ropivacaine 4 cc and 5% dextrose 6 cc) was injected into the catheter to reduce the initial postoperative pain. Then, an interscalene-PCA (AutoMed 3200; ACE Medical, Seoul, Korea) containing 0.3% ropivacaine 40 mL and 5% dextrose 60 mL (total 100 mL) was connected to the catheter line (basal infusion rate 2 mL/h; bolus 2 mL; lockout time 10 min). Both SSIB and IICA were performed by a single anesthesiologist.

#### 2.4.3. Postoperative Management

Pain was assessed by a VAS and measured at 4 time points: 6 h, 12 h, 24 h, and 48 h after ARCR. If the pain score was greater than 7 according to the VAS checklist, extra analgesics were given intravenously (i.v.), namely, fentanyl 50 mcg intravenously (i.v.) in the postanesthetic care unit (PACU), and then one of the three analgesics (pethidine 25 mg, tramadol 50 mg, and diclofenac 37.5 mg) in the ward in the following order: pethidine i.v., tramadol i.v., and diclofenac intramuscularly (i.m.). The total number of analgesics used as well as the number of each analgesic used were measured. While measuring the VAS score, nausea, vomiting, and dizziness were also checked.

### 2.5. Outcomes

The primary outcome was the comparison of pain severity at each time point between group D and group C, and the main time point was after 6 h. Secondary outcomes included the comparison of and correlations between duration of surgery, postoperative analgesic requirements, and VAS.

### 2.6. Power Analysis of Sample Size

There was a difference in the number of patients in the two groups. Therefore, a power analysis was performed as the power of pain score (VAS) was calculated. The standard deviation of the VAS at 12 h after ARCR was derived as 2.20 for group C and 2.85 for group D (Section 3.2). The non-inferiority threshold for power analysis was defined as 2 as usual [15,16]. The statistical power calculated using the Non-inferiority Test of Two Means using Differences of the PASS 2013 program was 98.763%, with alpha 5% in the number of samples in group C (*n* = 36) and group D (*n* = 94). Even when the non-inferiority threshold was assumed to be 1.5, the statistical power was calculated to be 88.379%.

### 2.7. Statistical Analysis

Differences in general characteristics between the two groups were compared using Chi-square test or Fisher’s exact test for categorical variables, and *t*-test or Wilcoxon rank sum test for continuous variables. Values are presented by numbers (percentages) for categorical variables and as the mean and standard deviation or the median with interquartile range for continuous variables. The VAS score at 6, 12, 24, and 48 h was compared between the groups using the Wilcoxon rank sum test. Regression analysis was performed to determine the differences in VAS score at each time point between the two groups after correcting for potential confounding variables. The number of postoperative and analgesic uses between the two groups was compared using Wilcoxon rank sum test. All statistical analyses were performed using SAS version 9.4, and *p* < 0.05 was considered to indicate statistical significance.

## 3. Results

### 3.1. Baseline Characteristics 

There were no differences in sex, age, body mass index (BMI), tear size or inserted anchors between the groups. The duration of surgery in group D was significantly longer than that in group C (Table 1).

### 3.2. Comparison of VAS Scores at Each Time Point between Group D and Group C

When comparing the VAS score at each time point between the two groups, group C showed a higher VAS score than group D at 6 h. However, there were no differences in the VAS score at 12, 24, and 48 h (Table 2).

### 3.3. Comparison of the Number of Postoperative Analgesic Uses between Group D and Group C

The number of times for total analgesics and fentanyl was significantly greater in group C, but there were no differences in the number of times the other analgesics were used between the groups (Table 3).

### 3.4. The Difference in VAS at Each Point Adjusted by Potential Compounders

When we considered confounding factors such as sex, duration of surgery, and total analgesics, group D had a significantly lower VAS than group C at 6 h. There were no significant differences at other time points except 6 h, though group D showed lower VAS scores at 48 h (Table 4).

### 3.5. Comparison of Side Effects Occurrence between the Groups

There was no difference in side effects such as dizziness, nausea, and vomiting between group D and group C (Table 5).

## 4. Discussion

ARCR results in a high degree of pain in the early postoperative period. Therefore, a number of multimodal pain management techniques have been proposed to relieve this pain [17,18]. Among these, nerve block is recommended over intravenous PCA (IV-PCA) because the latter, as well as opioid and inhalation anesthesia use, operation duration, and patient factors, are more likely to induce nausea and vomiting.

Regional nerve block using local anesthetics has been attempted via different routes, such as the interscalene, supraclavicular, and suprascapular routes and along the brachial plexus, and their effectiveness has been compared in numerous studies [19,20,21,22]. In addition to the nerve block site, there are various opinions on whether a single shot or continuous injection using a catheter is superior [2,23,24,25]. Continuous infusion using a catheter is effective for relieving pain, but it can prolong the length of hospital stay, induce a greater number of side effects, and must be monitored cautiously [25,26,27]. Single-shot nerve block is superior to indwelling catheter nerve block during the very early postoperative period (less than 2 h), but after 12 h, pain relief is superior with indwelling catheter nerve block [28]. In our study, initial postoperative analgesics in the very early period (<6 h) were used more frequently in group C than in group D. This suggests that the very early postoperative pain was larger in group C, similar to the findings of a previous study [28]. Regarding the duration of the operation, group D had a higher value than group C, but the VAS score was not higher in group D. This suggests that the operation time is not related to the increase in postoperative pain; in the case of SSIB, Dex can exert a synergic effect and could be a favorable choice for nerve block. In addition, with the use of Dex with SSIB, the VAS score more than 6 h after ARCR was not significantly different between the groups, which means that the analgesic effect could be continued for more than 6 h. Generally, immediately after ARCR, SSIB is effective due to the initial high concentration of local anesthetics. However, the efficacy gradually decreases, and rebound hyperalgesia occurs. On the other hand, even if a certain amount of the analgesics from IICA was given in bolus, the effect would not be substantial within 6 h due to its lower concentration than that used for SSIB and its continuous method of action for pain relief, but ultimately, the determination is made by operator preference in terms of the advantages and disadvantages of the type of nerve block [28]. Correspondingly, it is important to control pain after 6 h, and a way to maximize the effect of the nerve block is needed. In this study, there was no difference in the analgesic effect between SSIB and IICA even after 6 h, when the efficacy of SSIB starts decreasing. This means that SSIB showed a similar effect to IICA 6 h after ARCR, and the opportunity to extend the analgesic effect of SSIB can be increased by more than the existing effect duration by adding dexamethasone. Ultimately, this can reduce the amounts of initial analgesics administered and the related side effects caused. Except for patients requiring IICA for long-term pain control, it may be better to add perineural Dex for more efficient and higher quality pain management. Sometimes, there is a difference in the persistence of the drug effect depending on the dose of the local anesthetics and the skill with which the anesthetic techniques are performed. However, we suggest that there is no need to maintain IICA to overcome the shortcomings of SSIB according to our results, such as the similar VAS scores and postoperative analgesics used as well as the side effects that developed between the groups.

This study has some limitations. First, we did not calculate the sample size. The experimental group (group D) and control group (group C) were formed prospectively without unification of conditions through randomization. Second, the association of the analgesic use according to the exact onset time of pain was not suggested, and the doses of additional analgesics were not standardized by conversion to opioid equivalents. Third, although the order of administration of analgesics was determined, there were cases where the order was not performed at the discretion of the ward personnel. Therefore, it was focused on the evaluation based on the total amount of analgesics for pain.

However, based on the above results, in the next study, we will attempt to overcome these limitations and uncover a stronger basis for the use of SSIB over IICA.

## 5. Conclusions

ARCR produces severe pain during the early postoperative period. Although SSIB maintains a more effective analgesic effect than IICA in the early period, especially less than 6 h, it cannot relieve any rebound hyperalgesia, and the duration of the analgesic effect cannot be extended without the administration of an adjuvant. In our study, adding perineural dexamethasone to SSIB prolonged the analgesic effect. Therefore, combination of dexamethasone and SSIB might replace IICA for pain control even after operation.

## Figures and Tables

**Table 1 jcm-11-03409-t001:** Baseline characteristics.

	Total	Group D	Group C	
	*n* = 130	*n* = 94	*n* = 36	*p*-Value
Sex				0.075
Male	56 (43.08)	36 (38.30)	20 (55.56)	
Female	74 (56.92)	58 (61.70)	16 (44.44)	
Age				0.684 *
Mean (SD)	63.94 (6.60)	64.09 (6.23)	63.56 (7.57)	
Median (IQR)	63.50 (60.00, 68.00)	64.00 (60.00, 68.00)	62.50 (59.00, 67.00)	
BMI				0.658
Mean (SD)	24.73 (3.35)	24.85 (3.49)	24.43 (2.97)	
Median (IQR)	24.48 (22.23, 26.15)	24.59 (22.35, 26.26)	24.00 (22.07, 25.90)	
Tear size (AP, cm)				0.782
Mean (SD)	1.51 (0.94)	1.54 (0.97)	1.44 (0.88)	
Median (IQR)	1.15 (1.00, 2.00)	1.00 (1.00, 2.00)	1.50 (0.85, 1.85)	
Duration of surgery (min)				<0.001
Mean (SD)	95.62 (28.37)	101.09 (29.25)	81.33 (20.08)	
Median (IQR)	90.00 (75.00, 110.00)	95.00 (80.00, 110.00)	80.00 (67.00, 95.00)	
Inserted anchors (*n*)				0.316
Mean (SD)	2.38 (1.44)	2.47 (1.50)	2.14 (1.25)	
Median (IQR)	2.00 (1.00, 3.00)	2.00 (1.00, 3.00)	2.00 (1.00, 3.00)	

Numbers (percentages) for categorical variables. Mean (SD), median (IQR) for continuous variables. *p*-values were calculated using Chi-square or Fisher’s exact test. *: *t*-test; other variables were compared with the Wilcoxon rank sum test. SD: standard deviation; IQR: interquartile range. Group D: dexamethasone group; Group C: catheter group; min: minutes; *n*: number.

**Table 2 jcm-11-03409-t002:** Comparison of VAS scores at each time point between the groups.

Group	VAS Score (Mean ± SD)							
	at 6 h	*p* Value *	at 12 h	*p* Value	at 24 h	*p* Value	at 48 h	*p* Value
Group D	2.01 (2.32)	<0.001	3.94 (2.85)	0.863	4.00 (2.71)	0.391	2.74 (2.51)	0.378
Group C	4.22 (2.56)		3.92 (2.20)		3.47 (2.27)		3.11 (2.57)	

*: Wilcoxon rank sum test for continuous variables. Mean (SD): mean and standard deviation; * < 0.05. Group D: dexamethasone group; Group C: catheter group.

**Table 3 jcm-11-03409-t003:** Comparison of the number of postoperative analgesic uses between the two groups.

	Total (*n* = 130)	Group D (*n* = 94)	Group C (*n* = 36)	*p* Value *
no. of uses of fentanyl				<0.001
Mean (SD)	0.35 (0.66)	0.12 (0.38)	0.97 (0.81)	
Median (IQR)	0 (0, 1)	0 (0, 0)	1 (0, 2)	
no. of uses of pethidine				0.339
Mean (SD)	2.75 (1.59)	2.72 (1.72)	2.83 (1.21)	
Median (IQR)	3 (2, 3)	2 (2, 3)	3 (2, 3)	
no. of uses of tramadol				0.458
Mean (SD)	2.58 (2.13)	2.49 (2.02)	2.81 (2.41)	
Median (IQR)	2 (2, 4)	2 (1, 4)	2 (2, 4)	
no. of uses of diclofenac				0.547
Mean (SD)	0.99 (1.31)	0.99 (1.24)	1.00 (1.51)	
Median (IQR)	1 (0, 2)	1 (0, 2)	0 (0, 1)	
no. of total analgesics				0.033
Mean (SD)	6.68 (3.58)	6.32 (3.54)	7.61 (3.59)	
Median (IQR)	6.00 (4.00, 8.00)	5.00 (4.00, 8.00)	7.00 (5.00, 10.00)	

*: Wilcoxon rank sum test; <0.05 was considered statistically significant. SD: standard deviation; IQR (Q1, Q3): interquartile range; Group D: dexamethasone group; Group C: catheter group; no: number.

**Table 4 jcm-11-03409-t004:** Association between groups and VAS at each time point.

	Coeff.	S.E	*p*-Value *
Association between groups and VAS score at 6 h			
Group			
D	−2.084	0.489	<0.001
C	Ref		
Association between groups and VAS score at 12 h			
Group			
D	0.221	0.507	0.664
C	Ref		
Association between groups and VAS score at 24 h			
Group			
D	0.874	0.491	0.077
C	Ref		
Association between groups and VAS score at 48 h			
Group			
D	−0.167	0.489	0.738
C	Ref		

*: calculated using lineal regression analysis adjusted by sex, duration of surgery and total analgesics; Group D: dexamethasone group; Group C: catheter group.

**Table 5 jcm-11-03409-t005:** Comparison of side effects between the groups.

	Total	Group D	Group C	
	*n* = 130	*n* = 94	*n* = 36	*p*-Value
Dizziness				0.187
No	86 (66.15)	59 (62.77)	27 (75.00)	
Yes	44 (33.85)	35 (37.23)	9 (25.00)	
Nausea				0.104
No	91 (70.00)	62 (65.96)	29 (80.56)	
Yes	39 (30.00)	32 (34.04)	7 (19.44)	
Vomiting				>0.999 †
No	113 (86.92)	82 (87.23)	31 (86.11)	
Yes	17 (13.08)	12 (12.77)	5 (13.89)	

† *p*-values calculated using the chi-square test, Fisher’s exact test. Values are presented as numbers (percentages) for categorical variables.

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
