# Peer review of "Is There a Difference between Perineural Dexamethasone with Single-Shot Interscalene Block (SSIB) and Interscalene Indwelling Catheter Analgesia (IICA) for Early Pain after Arthroscopic Rotator Cuff Repair? A Pilot Study"

_jcm, 2022, doi:10.3390/jcm11123409_

Round 1
Reviewer 1 Report
In this prospective study, the authors investigated single-shot interscalene block (SSIB) with the addition of dexamethasone and compared it with continuous interscalene indwelling catheter analgesia (IICA) in the context of arthroscopic rotator cuff repair. They focused mainly on the early postoperative period and demonstrated that SSIB with dexamethasone is superior to IICA as it reduced rebound hyperalgesia by ensuring lower VAS scores 6 hours after rotator cuff repair and lower use of fentanyl in the Postanesthesia Care Unit as compared to IICA. After 6 hours, there were not significant differences and analgesic requests between the two groups, suggesting a prolonged action by the addition of dexamethasone to the local anesthetic mixture. Therefore, the authors concluded that adding dexamethasone may offer advantages by ensuring a prolonged analgesic effect, obviating the need for using indwelling catheters.
My comments are as follows:
- Did the authors register their study in a database like clinicaltrials.gov before recruiting patients? This is mandatory for clinical studies nowadays. Please comment…
- It appears that the authors did not perform a power analysis to define the necessary sample size for their study. In order to do so, they should have set a primary outcome measure (for example pain scores or analgesic consumption postoperatively) and base the required sample size on this outcome measure. Since the absence of sample size calculation appears to be the case, the authors should clearly acknowledge this as a limitation.
- Why is there a discrepancy in the number of patients allocate in the two groups? I would expect a similar number of patients per group based on the allocation methodology a described by the authors
- Please rewrite phrases between lines 42-46 and 56-59 because they are unclear to the readers
- Line 84: correct to: “in which they underwent…”
- line 101: please refer to the dose of dexamethasone used in mg and not to the volume of solution in ml
- It appears form the Anesthetic Methods description that both cohorts received a bolus of local anesthetic . Please comment on the Discussion section
- According to the authors’ description, one of the three analgesics (pethidine 25 mg, 115 tramadol 50 mg and diclofenac 37.5 mg) was administered during the postoperative period in case of postoperative pain. I believe that postoperative analgesia should have been standardized in order to achieve meaningful comparisons between the groups. It would have made sense to convert to opioid equivalents and perform the comparisons after conversion. Was the selection of each specific analgesic administered at the discretion of the ward personnel? Please comment
- It would be more appropriate to compare postoperative analgesic requests between the two groups by using a non-parametric method and describe in terms of median and interquartile range (Table 3).
- Line 191: correct to: “ the occurrence of side effects induced by catheter placement is more common…”
- line 197: correct to: “ the very early postoperative pain was more intense…”
Finally, there are a few grammatical and syntax mistakes in the manuscript , which would greatly benefit from a language editing service to correct them before further consideration.
Author Response
Thank you very much for reviewing my article and providing detailed comments.
I answered your comments as follows; and the parts you pointed out has been corrected and marked in blue through the article.
We checked and corrected all the parts you mentioned, and in general, the tables and figures have been modified to express the contents concisely. In order to convey the precise message of this article, parts other than those you pointed out are marked in green.

Reviewer 2 Report
Dear authors,
you presented a clinical study on a topic of particular interest in the field of anesthesiology. Your results support the concept of a single-shot ISB suitable for the ambulatory outpatient setting and excluding the risks from indwelling catheters. While I see this topic of high importance and impacting the clinical practice I also see a few things that should be improved before publication.
Major comment: You miss the sample size calculation. If you planned your study to find a significant difference it should be based on this. If not it has to be clearly stated and even your title should be updated to state that your study was a pilot. This has to be described in detail in the methods section.
Minor comments:
1) Introduction, line 42-43 -> citation needed.
2) Participants, line 75 -> cognitive dysfunction, and so on, ... - what did you mean by "so on"? You have to be precise. This sounds to me quite unscientific.
3) Operative techniques: Were the anesthesia/peripheral nerve blocs done also by one anesthesiologist? If not it lowers the power of your study a bit.
4) Anesthetic methods: Here I lack dosing of all the agents used and strategy and description of the airway management. What was the exact dose in mg of dexamethasone used? 4 mg?
5) Outcomes: In primary outcomes use a singular form since you mention only one.
6) Statistical analysis: Improve the wording: Data were analyzed in comparison between the two groups... What test did you use for normality data testing?
7) Results: line 133: improve wording: operating time -> duration of the surgery. Line 143 This is shown...
8) Table 2: I can imagine this table without p values, it would be more clear and brief which is a scientific target in writing, instead of p values I would welcome just asterik in case of statistical significancy and particualr footnote. Use spaces between the symbols.
9) Figure 1: The graph picture is not of sufficient quality, I guess 600 DPI would look much better.
10) Line 169: What do you mean by previous studies? In results there shouldbe no comparison.
11) Discussion: 192: improve wording.
12) I do not see this as a limitation, you just chose one way af application –
perineural. More importantly, the sample size was not calculated as in the previous comment.
13) Line 264: citations should contain up to 6 authors with et al.
Author Response
Thank you very much for reviewing my article and providing detailed comments.I answered your comments as follows; and the parts you pointed out has been corrected and marked in red through the article.
We checked and corrected all the parts you mentioned, and in general,the tables and figures have been modified to express the contents concisely. In order to convey the precise message of this article, parts other than those you pointed out are marked in green.

Round 2
Reviewer 2 Report
Dear authors,
your manuscript was improved significantly. Now it seems to me neat.
Although I have one small suggestion, you described the calculation of the sample size and power analysis in the discussion. It should be in the methods section. Please change this.
I am still unsatisfied with your english. Native speaker correction would be valuable. Please consider.
Author Response
Thank you very much for reviewing my article and showing positive response to it.
Changed sentences(paragraphs) were marked in red through the article.
Response to Reveiwer 2 2nd Comments
Thank you very much for reviewing my article and showing positive response to it.
Changed sentences (paragraphs) were marked in red through the article.
Although I have one small suggestion, you described the calculation of the sample size and power analysis in the discussion. It should be in the methods section. Please change this.
Response: I rearranged power analysis description from ‘study limitation in discussion section’ to ‘materials and methods’ section as you mentioned
I am still unsatisfied with your English. Native speaker correction would be valuable. Please consider.
Response: I am very sorry that my English is not satisfactory to you. I re-uploaded as a revised manuscript by a native speaking professional translator with certification.